# Assessing the Toxicity of *Lagocephalus sceleratus* Pufferfish from the Southeastern Aegean Sea and the Relationship of Tetrodotoxin with Gonadal Hormones

**DOI:** 10.3390/md21100520

**Published:** 2023-09-29

**Authors:** Thekla I. Anastasiou, Eirini Kagiampaki, Gerasimos Kondylatos, Anastasios Tselepides, Panagiota Peristeraki, Manolis Mandalakis

**Affiliations:** 1Hellenic Centre for Marine Research (HCMR), Institute of Marine Biology, Biotechnology and Aquaculture, 71500 Heraklion, Greece; theanast@hcmr.gr (T.I.A.); e.kagiampaki@hcmr.gr (E.K.); 2Department of Biology, University of Crete, 70013 Heraklion, Greece; tselepidis@uoc.gr; 3Hellenic Centre for Marine Research (HCMR), Hydrobiological Station of Rhodes, 85131 Rhodes, Greece; gkondylatos@hcmr.gr; 4Hellenic Centre for Marine Research (HCMR), Institute of Marine Biological Resources and Inland Waters, 71500 Heraklion, Greece; notap@hcmr.gr

**Keywords:** biotoxins, *Lagocephalus sceleratus*, silver-cheeked toadfish, Lessepsian species, pufferfish toxicity, steroid hormones, Eastern Mediterranean

## Abstract

Given the dramatic increase in the *L. sceleratus* population in the southeastern Aegean Sea, there is growing interest in assessing the toxicity of this pufferfish and the factors controlling its tetrodotoxin (TTX) content. In the present study, liver, gonads, muscle and skin of 37 *L. sceleratus* specimens collected during May and June 2021 from the island of Rhodes, Greece, were subjected to multi-analyte profiling using liquid chromatography-tandem mass spectrometry (LC-MS/MS) in order to quantitate TTX and evaluate whether this biotoxin interrelates with hormones. TTX and its analogues 4-epiTTX, 11-deoxyTTX, 11-norTTX-6-ol, 4,9-anhydroTTX and 5,11/6,11-dideoxyTTX were detected in all tissue types. Liver and gonads were the most toxic tissues, with the highest TTX concentrations being observed in the ovaries of female specimens. Only 22% of the analyzed muscle samples were non-toxic according to the Japanese toxicity threshold (2.2 μg TTX eq g^−1^), confirming the high poisoning risk from the inadvertent consumption of this species. Four steroid hormones (i.e., cortisol, testosterone, androstenedione and β-estradiol) and the gonadotropin-releasing hormone (GnRH) were detected in the gonads. Androstenedione dominated in female specimens, while GnRH was more abundant in males. A positive correlation of TTX and its analogues with β-estradiol was observed. However, a model incorporating sex rather than β-estradiol as the independent variable proven to be more efficient in predicting TTX concentration, implying that other sex-related characteristics are more important than specific hormone-regulated processes.

## 1. Introduction

*Lagocephalus sceleratus* (Gmelin, 1789) is among the 13 pufferfish species recorded so far in the Mediterranean Sea [1] and the most noxious invasive fish in Greece. This is a member of the Tetraodontidae family with Indo-Pacific origin, which entered the Mediterranean Sea through the Suez Canal. The widespread distribution and high toxicity of this species has been documented in several studies, and 18 years after its first appearance in Northern Crete, Greece [2], it still poses a threat to local biodiversity, fisheries and citizens, while similar issues are faced in neighboring countries [3,4,5]. Intriguingly, the general public (natives and tourists) is still unaware of this poisonous fish, even though this issue has received extensive publicity in local media. Moreover, the great majority of people are unable to recognize it, despite its distinctive external morphological characteristics (i.e., head, skin, teeth), thus increasing the risk of poisoning from its consumption.

This silver-cheeked toadfish is known for its strong bite force and its high body content of tetrodotoxin (TTX), one of the most lethal neurotoxins found in nature. The assessment of TTX levels in its tissues has been the subject of several studies [6,7,8], while some investigations have focused on the valorization of TTX isolated from fish of the Tetraodontidae family for medical or pharmaceutical purposes [9,10,11,12]. Moreover, there is ongoing research on compounds of high commercial interest that can be obtained from this species, such as collagen, omega-3- and omega-6 fatty acids, and their concomitant use in cosmetology and food industry (e.g., EXPLIAS project) [13]. Reports on the utilization of *L. sceleratus* body parts are scarce and limited to the studies of Doğdu et al. [14,15], which were targeted on skin and teeth valorization.

Besides TTX, there are about 30 compounds with analogous chemical structure and varying toxicity that are difficult to identify and quantify, mainly due to the lack of commercially available standards. A number of studies have tried to find a correlation between the concentration of the toxin (TTX alone or in combination with its analogues) in tissue samples and the morphological/biological characteristics of pufferfish or some key environmental parameters [6,16]. So far, these have had only limited success, suggesting that the indirect prediction of toxicity levels in pufferfish individuals is much more complicated than simply using fish/environment-related data. Nevertheless, the Japanese regulatory limit of 10 mouse units (MU) of TTX eq per gram of fish flesh (corresponding to 2.2 μg TTX eq g^−1^) [17,18] is typically used as a toxicity threshold for all different pufferfish species. Meanwhile, the trading of Tetraodontidae pufferfish is prohibited in the EU, but a European safety limit for TTX has not yet been established, even though the same toxin is occasionally found in edible organisms, such as bivalves [19,20]. Upon request by the European Commission, the European Food Safety Authority (EFSA) concluded that an amount lower than 0.044 μg TTX eq g^−1^ shellfish meat is not anticipated to pose a risk to human health [21].

Although the results of previous studies regarding the effect of various parameters on pufferfish TTX levels are frequently inconsistent or contradictory, fish sex is increasingly recognized as playing a key role. In general, the concentration of TTX has been shown to be considerably higher in female than in male pufferfish [6,22,23], but the actual reason for this differentiation remains unclear. Besides serving as a direct defense against predators, a contribution to the spawning process has been postulated and maternal TTX has been suggested to protect pufferfish eggs and larvae [24,25]. On the other hand, TTX has been suggested to act as a male-attracting sex pheromone [26], a behavior that is typically mediated by fish steroids, prostaglandins and their metabolites [27].

Considering the inferred association of TTX with sexual differentiation/maturation and reproduction processes in pufferfish, we were intrigued to investigate the potential relationship of this toxin with sex steroid hormones. The latter are regarded as the chemical messengers regulating a vast number of physiological processes, but they are also molecules with a leading role in the development of sexual characteristics among males and females [28,29]. To date, there is only a single study on the interaction of TTX with steroid hormones, where specimens of the pufferfish *Takifugu rubripes* with high TTX content exhibited lower plasma cortisol concentration [30]. However, cortisol is mostly considered a stress-related hormone and it is not included among the most relevant sex steroid hormones (i.e., androgens and estrogens).

In this study, the levels of TTX and several of its analogues were measured in different body parts (gonads, liver, muscle and skin) of *L. sceleratus* specimens collected from Rhodes Island, southeastern Greece, while further measurements for several steroid hormones were performed in the gonad tissues. Besides assessing the overall toxicity of this invasive pufferfish, our main objective was to evaluate whether the toxicity levels were related with sex hormones. To the best of our knowledge, this is the first study investigating the hormonal profile of this toxic Lessepsian migrant in connection with tetrodotoxin content.

## 2. Results and Discussion

### 2.1. Concentration Levels of TTX and Related Analogues

Tetrodotoxin and its analogues 4-epiTTX, 4,9-anhydroTTX, 11-norTTX-6-ol, 11-deoxyTTX and 5,11/6,11-dideoxyTTX were detected in all types of *L. sceleratus* tissues (i.e., gonads, liver, muscle and skin), whereas 5,6,11-trideoxyTTX was detectable only in liver and gonad samples. A representative LC-MS/MS chromatogram of the analytes under investigation is shown in Figure 1. The detection rate of tetrodotoxin, 11-norTTX-6-ol, 4-epiTTX and 11-deoxyTTX among the samples analyzed was 100%, while 4,9-anhydroTTX and 5,11/6,11-dideoxyTTX were detected in 93% and 99% of the samples, respectively. On the contrary, 5,6,11-trideoxyTTX was detected only in 10% of the samples (i.e., 14 out of the 145) and at concentrations that did not exceed 0.08 μg g^−1^. For this reason, the specific TTX-analogue was excluded from further analysis

The average concentration and the range of values measured for tetrodotoxin and its analogues in the different tissue types of *L. sceleratus* are summarized in Table 1. Tetrodotoxin was the most abundant compound in all tissues, with an average concentration ranging from 5.6 ± 0.9 μg g^−1^ in the skin up to 43.7 ± 7.7 μg g^−1^ in the liver. The second most abundant analogue was 11-norTTX-6-ol (2.3 ± 0.4 [muscle, skin]–31.8 ± 5.6 μg g^−1^ [liver]), followed by 4-epiTTX (0.3 ± 0.05 [muscle]–5.4 ± 1.6 μg g^−1^ [liver]) and 11-deoxyTTX (0.3 ± 0.04 [muscle, skin]–4.5 ± 0.8 μg g^−1^ [liver]). Although 4,9-anhydroTTX was below detection limit in 10 samples, the mean concentrations measured for this TTX-analogue (0.2 ± 0.03 [muscle]–3.7 ± 0.7 μg g^−1^ [liver]) were similar to those of 4-epiTTX and 11-deoxyTTX. On the contrary, the levels of 5,11/6,11-dideoxyTTX (0.02 ± 0.004 [muscle]–0.4 ± 0.1 μg g^−1^ [liver]) were almost an order of magnitude lower compared to all other TTX-analogues.

Due to the higher toxicity of TTX compared to its analogues [21], it is usually the only compound measured in pufferfish and other marine organisms, while the TTX-analogues are rarely investigated. Recently, the predominance of TTX was observed in all tissues of *L. sceleratus* specimens collected from the northern and southern Cretan Sea [6], but the profile of TTX-analogues (11-deoxyTTX > 4,9-anhydroTTX ≈ 11-norTTX-6-ol > 4-epiTTX ≈ 5,11/6,11-dideoxyTTX) demonstrated considerable differences compared to our study. Rodriguez et al. [31] studied the toxicity of *L. sceleratus* specimens from the island of Rhodes almost a decade ago. Strangely, 5,6,11-trideoxyTTX analogues dominated in all of the tissues analyzed (i.e., skin, muscle, gastrointestinal tract, liver, gonads), followed by 11-norTTX-6(S)-ol and 11-deoxyTTX, while much lower levels of TTX were recorded. A prevalence of 5,6,11-trideoxyTTX in the liver and gastrointestinal tract was also reported by Bane et al. [32], whereas 11-deoxy and 11-norTTX-6(S)-ol were the next most abundant toxins. Overall, the profile of TTX and its analogues seems to be highly variable among different studies, but these discrepancies could be partly explained by the dissimilarities in the analytical methods applied and the distinct characteristics of the pufferfish populations investigated (e.g., sex ratio, maturation stage, sampling area and season, sampling frequency, size of individuals and number of samples). More importantly, we consider that the lack of a commercially available Certified Reference Material for TTX-analogues leads to limited accuracy in their quantification and the LC-MS/MS concentration data reported in the literature have been practically based on semiquantitative analytical approaches.

In general, liver and gonads exhibited the highest toxin levels, with the total concentration of TTX and its analogues (∑TTX) approaching 89.4 ± 15.5 μg g^−1^ and 54.6 ± 8.9 μg g^−1^, respectively. Much lower levels were observed in muscle and skin, with ∑TTX values reaching 9.4 ± 1.3 μg g^−1^ and 8.9 ± 1.3 μg g^−1^, respectively. Regardless of tissue type, the contribution of tetrodotoxin to ∑TTX exceeded 53%. The pronounced presence of TTX in the liver and gonads of *L. sceleratus* has already been observed for specimens collected from Greece [6,33], Cyprus [7] and Turkey [16,34].

A further comparison between female, male and immature specimens (i.e., individuals that did not have fully developed reproductive organs and could not be categorized as male or female) was performed, focusing on TTX concentrations of the different tissues. This investigation revealed that female liver and gonads had an almost 5 times higher TTX concentration compared to muscle and skin. A similar pattern was observed for males, although TTX levels in gonads were only 3-fold higher than those in muscle and skin. With regard to immature specimens, the stronger presence of TTX in gonads than any other tissue type was their main characteristic. In general, the concentration of TTX in muscle and skin samples was almost identical and this homogeneity was consistently evident in all three sex groups. More interestingly, the TTX levels in females were considerably higher than those measured in immature specimens, although the difference in gonads was less pronounced compared to the other tissue types (Figure 2). Furthermore, the TTX content in gonad, muscle and skin of females was nearly three times higher than those of males, whereas comparable concentrations were recorded in the liver tissues of both sex groups.

Although there are no studies investigating the underlying mechanisms of this gender-related TTX accumulation in *L. sceleratus*, it has been reported that male reproductive organs are less effective in TTX storage [22]. More importantly, research on different pufferfish species has shown that high toxicity in female tissues may be associated with the enhancement of reproduction success and their offspring’s’ survival. More specifically, studies on *Takifugu niphobles* and *T. rubripes* have shown that TTX in female pufferfish acts as a pheromone, with male pufferfish being attracted through their olfactory system by low toxin levels [26,35]. Moreover, Ikeda et al. [23] and Itoi et al. [22] observed that the ovaries of *T. poecilonotus* and *T. niphobles* exhibited higher toxin levels than testes during spawning season, while Itoi et al. [25] and Gao et al. [24] reported that females belonging to different *Takifugu* species transferred TTX from the ovaries to their eggs and larvae after spawning, possibly as a protecting mechanism against predation. Taking all these findings into consideration, the higher TTX content recorded in most of the female tissues analyzed in the present study could be attributed, at least partly, to the sampling period (May–June), which corresponded to the beginning of the spawning season for *L. sceleratus* [1,3,36].

Regarding the high TTX content detected in *L. sceleratus* liver, this could be related to hepatic uptake. Indeed, several studies have investigated long-term (oral administration) or short-term (intramuscular or intravenous administration) distribution and accumulation of TTX in the tissues of different pufferfish species. According to this research, tetrodotoxin is reported to be initially accumulated at high concentrations in the liver, and then transferred to the skin (especially in immature or male pufferfish), or in the ovaries of females, through blood circulation [37,38,39,40]. Although these findings regarding the intrabody transfer of TTX have been based only on T. rubripes, which is an edible species [41], similar circulation mechanisms are expected to control toxin accumulation in other pufferfish species. With regard to *L. sceleratus*, this aspect has been poorly investigated and most studies simply focus on its high toxicity. Thus, further research is warranted to elucidate the underlying mechanisms of selective TTX accumulation and transportation in *L. sceleratus* internal organs.

### 2.2. TTX-based Toxicity of L. sceleratus

Total toxicity of *L. sceleratus* tissues is defined as the sum of the individual toxicities of TTX and its analogues, expressed in micrograms of TTX equivalents per gram of tissue (μg TTX eq g^−1^). In practice, this is calculated by converting the concentrations data to TTX equivalents using the relative potency (RP) of each compound, as proposed by Knutsen et al. [21]. Given its greater concentration and toxic potency, TTX was by far the largest contributor to the total toxicity calculated for *L. sceleratus* species (Table 2). As expected, the overall toxicity of liver and gonads was much higher compared to muscle and skin tissues.

Total toxicities calculated for the individual tissue samples were further used to categorize them as toxic or non-toxic, based on the Japanese safety consumption limit of 2.2 μg TTX eq g^−1^. This assessment revealed that only 15% of all tissue samples analyzed in the present study were non-toxic (Figure 3). Interestingly, only 22% of the muscle samples analyzed, corresponding to two female and six male specimens, were regarded as “risk-free”. This percentage is much lower compared to our previous results on *L. sceleratus* from the Cretan and Libyan Sea, where almost half (i.e., 48%) of the total muscle tissues analyzed were non-toxic [6]. Based on the literature, the sampling season has been shown to play an important role on TTX concentrations [8,36]. To verify whether this parameter was responsible for the observed difference in the percentage of non-toxic samples between the present and our former study, a comparison based on specimens collected during the same season (i.e., spring) was also carried out. Even after using season-specific data, the percentage of non-toxic samples calculated from Christidis et al. [6] was only slightly changed (i.e., 52%) and remained much higher compared to our results. Besides sampling season, Christidis et al. [6] demonstrated that TTX levels in *L. sceleratus* are strongly affected by the sampling area, and this finding was attributed to differences in fish diet composition and/or habitat conditions. Indeed, several studies suggest tetrodotoxin accumulation via the oral route, as experiments on cultured pufferfish with TTX-free diet have shown that fish become toxic after ingestion of TTX and TTX-bearing organisms like gastropods or other toxic pufferfish [42,43,44,45]. Moreover, environmental conditions such as salinity, temperature and water depth have been suggested as possible factors that affect TTX production by bacteria and ΤΤΧ presence in host organisms [19,46,47,48]. Considering the above, the increased toxicity observed for *L. sceleratus* in the present study was more likely area-specific and resulted from the dietary resources available and the particular physicochemical conditions prevailing in the pufferfish habitats of Rhodes Island.

The Ministry of Health, Labour and Welfare in Japan has not only established safety limits for TTX in seafood, but also developed an official database where all pufferfish species recorded in the country are illustrated with colour photos, along with information on their external features (skin, length) and edible parts [49,50]. Meanwhile, in Europe, no legislation is in force for this biotoxin apart from the ban of pufferfish from the market (Regulation (EC) No 853/2004, Commission Implementing Regulation (EU) 2019/627) [51,52], despite the wide distribution of *L. sceleratus* in the Eastern Mediterranean and the numerous studies on the presence of TTX in pufferfish and molluscs from adjacent areas [20,53,54]. In 2017, the European Food Safety Authority (EFSA) proposed a safety limit of 0.044 μg TTX eq g^−1^ for shellfish meat [21] within the European Union, but this has not been validated for pufferfish. The specific limit is much more conservative (i.e., two orders of magnitude lower) compared to the one set in Japan, and it is worth stressing that the TTX levels in all samples analyzed in the present study greatly exceeded the EFSA limit. Considering the dramatic increase in *L. sceleratus* in the Greek seas, along with the high risk of poisoning due to accidental fish consumption, communication actions should be intensified to increase awareness in local communities and targeted measures should be taken to help reduce the populations of this invasive pufferfish.

### 2.3. Hormones in L. sceleratus Gonads

Gonads of female, male and immature *L. sceleratus* specimens were subjected to chemical analysis in an attempt to investigate their hormonal profile and its relationship with TTX levels. Among the 13 hormones targeted in the present study, only 4 of them, belonging to glucocorticoids (cortisol), androgens (testosterone, androstenedione) and estrogens (β-estradiol), along with the gonadotropin-releasing hormone (GnRH), were detected in the reproductive organs of this species.

Cortisol and testosterone were detected in all 34 gonad samples analyzed, at concentrations ranging from 0.001 to 0.32 μg g^−1^ and 0.0003 to 0.05 μg g^−1^, respectively (Table 3). The detection rate of GnRH and androstenedione was 97%, as each of these hormones was found below the detection limit in one male sample, and their abundancies varied from 0.03 to 0.91 μg g^−1^ and 0.001 to 2.40 μg g^−1^, respectively. The only determined estrogen, β-estradiol, was identified at concentrations varying from 0.001 to 0.06 μg g^−1^ in 76% of the samples analyzed. Interestingly, all eight non-detects of β-estradiol corresponded to male specimens.

In terms of average concentration, the most abundant hormone was androstenedione (0.23 ± 0.08 μg g^−1^), followed by GnRH (0.11 ± 0.03 μg g^−1^) and cortisol (0.06 ± 0.01 μg g^−1^) (Table 3). The corresponding values of β-estradiol and testosterone were considerably lower and equaled 0.020 ± 0.004 μg g^−1^ and 0.010 ± 0.002 μg g^−1^, respectively. The levels of hormones were further investigated in female, male and immature *L. sceleratus* specimens separately. Results showed that androstenedione dominated in female gonads, with an average concentration of 0.6 ± 0.2 μg g^−1^ (versus 0.05 ± 0.03 μg g^−1^ in males), while GnRH was more abundant in male gonads, at a mean concentration of 0.2 ± 0.1 μg g^−1^ (versus 0.05 ± 0.01 μg g^−1^ in females), (Table 4, Figure 4). In immature specimens, none of the hormones prevailed, as the mean concentrations of GnRH, cortisol, β-estradiol and androstenedione had similar values, ranging from 0.03 ± 0.02 to 0.06 ± 0.04 μg g^−1^.

Androstenedione is a sex steroid hormone that acts as pheromone in male goldfish by promoting an aggressive behavior which allows the demonstration of supremacy against subordinates [55,56]. Moreover, androgens together with GnRH are reported to be associated with territorial behavior in male fish [29]. However, androstenedione does not trigger the same response when it is released from females. More specifically, the secretion of this androgen from female goldfish is perceived by males as a signal that prevents courtship and reproduction with fish that are not in the ovulation phase (i.e., prior to spawning) [55,56]. Unfortunately, the lack of information regarding the maturity stage of *L. sceleratus* specimens in the present study impedes from reaching an interpretation regarding the predominance of specific hormones within each sex.

So far, there are limited studies investigating the presence of hormones in pufferfish. In particular, cortisol levels were determined as a response to TTX administration in *Takifugu rubripes* juveniles [30], while the effect of β-estradiol on the differentiation of gonads was examined in males of the same species [57]. Moreover, GnRH genes and a possible role of β-estradiol, testosterone and cortisol on their expression in *Takifugu niphobles specimens* were also investigated [58]. To the best of our knowledge, this is the first report on the hormonal profile of *L. sceleratus*, and therefore, further research is needed to gain more insights about the actual role of hormones in the physiology and behavior of this Lessepsian migrant.

The concentration results were further scrutinized to determine any potential correlations of the detected hormones with sex and/or the external fish characteristics (Table 5). Results revealed statistically significant positive correlation of β-estradiol, testosterone and androstenedione with sex (Pearson’s *r* = 0.66, *r* = 0.56 and *r* = 0.54, respectively; *p* < 0.002), implying higher levels of those hormones in female specimens, while GnRH was shown to be positively correlated with length and weight (Pearson’s *r* = 0.37 and *r* = 0.40, respectively; *p* < 0.03). Intriguingly, β-estradiol was the only hormone that showed a statistically significant positive correlation with TTX (Pearson’s *r* = 0.56; *p* = 0.001) and its analogues (Pearson’s *r* = 0.51–0.73, *p* < 0.002), whilst the rest of the hormones did not show any statistically significant correlation with either TTX or TTX-analogues (*p* > 0.05).

Furthermore, stepwise regression was applied to identify the subset of variables among those investigated (i.e., macroscopic fish characteristics and gonadal hormones levels) that best predicted TTX levels in gonads, as well as to assess the relative importance of those variables (Table 6). This analysis provided a statistically significant model (*p* < 0.001) that included only two predictor variables (i.e., fish sex and weight) and explained 62% of the variance in TTX concentration data. More specifically, higher TTX levels were predicted for samples of lower weight and of female sex. Based on the standardized coefficients of the two variables, sex (β = 0.58) had a considerably stronger effect on TTX concentration compared to weight (β = −0.42). Moreover, the calculated semi-partial correlations suggested that the unique contributions of sex and weight on the variance of TTX concentration were 33% and 17%, respectively.

After further examination, it was revealed that none of the hormones could enter into the model equation as long as fish sex was included among the predictor variables. By excluding sex from stepwise regression, a model incorporating fish weight and gonadal β-estradiol concentration as predictors was obtained. This second version was also characterized by high statistical significance (*p* < 0.001), but it explained only 46% of the variance in TTX concentration. In analogy to the previous model, higher TTX levels were predicted for pufferfish individuals having lower weight and higher β-estradiol concentration. In addition, the latter variable was found to have a greater effect on TTX concentration (β = 0.49 and −0.35 for β-estradiol concentration and fish weight, respectively). Nevertheless, the percentage of the variance in TTX concentration that could be uniquely attributed to β-estradiol (22%) and fish weight (11%) was much lower compared to the variables of the first model. Based on these results, sex is definitely a better predictor of TTX levels than β-estradiol concentration, implying that the presence of the toxin in *L. sceleratus* gonads is more likely affected by the general sex-related characteristics of the pufferfish rather than by specific hormone-controlled processes.

## 3. Materials and Methods

### 3.1. TTX Analysis

Specimens belonging to the species *Lagocephalus sceleratus* were caught along Rhodes Island, southeastern Greece, between May and June 2021. Total fish length and weight were measured for each one of the 37 individuals, while gender was identified after fish dissection (Table 7). Thereafter, muscle, skin, liver and gonads were removed and kept at −20 °C until chemical analysis. With the exception of three gonads of very low mass (i.e., <0.15 g), a total of 145 samples were analyzed for tetrodotoxin and its analogues, namely 4-epiTTX, 11-deoxyTTX, 11-norTTX-6-ol, 4,9-anhydroTTX, 5,11/6,11-dideoxyTTX and 5,6,11-trideoxyTTX. The concentrations of TTX and TTX-analogues from each sample were summed, and the average total concentration (∑TTX) was calculated for each tissue type separately.

The methodology described by Christidis et al. [3] was used for the analysis of TTX and TTX-analogues after minor modifications. More specifically, voglibose (Sigma-Aldrich GmbH, Germany; ≥97% purity) was used as an internal standard instead of N-methyl-D-glucamine. Furthermore, the extraction of TTX from tissue samples was performed in a single round (instead of two consecutive extraction rounds), as this proved sufficient for extracting over 93% of TTX. Regarding TTX separation via Solid Phase Extraction (SPE), custom-made cartridges consisting of 1-mL pipette tips plugged with wool were replaced by single fritted microcartridges (Kinesis-TELOS empty wells of 600 μL with polyethylene frits of 20 μm porosity; Cole Parmer, Wertheim, Germany) that were filled with 10 mg of polymer-based sorbent material (Strata-X-C 33 μm polymeric strong cation, Phenomenex, Aschaffenburg, Germany). All other extraction and detection parameters remained unchanged.

### 3.2. Hormone Analysis

Besides TTX determination, the levels of steroid hormones were also measured in *L. sceleratus* specimens using the method of Papadaki et al. [59] with some modifications. Due to the fact that fish blood sampling was not possible, the analysis of hormones had to be made on the available tissue samples. Considering that gonads exhibit much higher levels of TTX than other body parts [3] and they represent the primary organs for steroidogenesis, the specific type of tissue was selected for hormones measurements. Prior to sample extraction, several parameters including solvent type and volume, mass of extracted tissue and number of extractions were optimized to ensure quantitative recovery (i.e., >95%) for the analyzed hormones. Following the optimized extraction procedure, approximately 0.2 g of minced gonads were placed in 2 mL Safe-Lock tubes (Eppendorf; Hamburg, Germany) containing two 5 mm stainless steel beads (Qiagen; Hilden, Germany) and 750 μL of acetonitrile. Disruption of tissue samples was performed in a TissueLyser II (Retsch, Qiagen; Hilden, Germany) set at 25 Hz for 10 min, followed by centrifugation at 13,800 g for 10 min. The supernatants were collected in amber glass vials and the extraction procedure was repeated once more with 750 μL of fresh solvent. The final extracts were spiked with 10 μL of a 13C-labelled hormones mix containing cortisol (2,3,4-13C3; 83.4 pg μL^−1^), estradiol (3,4-13C2; 79.9 pg μL^−1^), testosterone (2,3,4-13C3; 20.7 pg μL^−1^) and progesterone (2,3,4-13C3; 9.8 pg μL^−1^) as internal standards (Cambridge Isotope Laboratories, Inc., USA; ≥98% purity). Samples were evaporated to dryness (EZ-2 centrifugal evaporator, Genevac Ltd.; Ipswich, United Kingdom) at 35 °C for 2.5 h and reconstituted in 400 μL of 20% methanol. Subsequently, hormones separation was performed using the SPE method previously described for plasma samples by Papadaki et al. [59]. In brief, microcartridges dry-packed with 10 mg of C18 sorbent (Strata-X 33 μm polymeric reversed phase; Phenomenex, Aschaffenburg Germany) were initially conditioned with 500 μL of methanol and 500 μL of water and, thereafter, the reconstituted samples were loaded. After applying two washing steps (500 μL of water and 400 μL of methanol 40% *v/v*), hormones were selectively eluted with 500 μL of methanol. Eluates were collected in amber glass vials, spiked with 10 μL of N,N dimethyl-L-phenylalanine (0.2 ng μL^−1^) as a recovery standard, evaporated to dryness and redissolved in 200 μL of methanol. A total of 13 hormones (adrenosterone, aldosterone, 4-androstene-3,17-dione, β-estradiol, cortisol, cortisone, estriol, estrone, GnRH, progesterone, testosterone, 11-ketotestosterone, 17α,20β-dihydroxy-4-pregnen-3-one) were analyzed with LC-MS/MS using previously optimized instrument parameters [59].

### 3.3. Quality Control and Assurance

Internal standard calibration curves were generated for TTX and its analogues by analyzing a series of seven standard solutions. The latter were prepared using a certified reference standard of tetrodotoxin that was purchased from CIFGA S.A. Laboratorio, Spain (CRM-03-TTXs; containing 21 ng μL^−1^ of TTX and trace levels of TTX-analogues). The calibration curve of TTX was linear at the concentration range 0.001–4.9 ng μL^−1^ and the same was evident for all TTX-analogues at concentrations ranging from 0.1 pg μL^−1^ to 1.3 ng μL^−1^. For all analytes, the regression coefficient R^2^ of the calibration curve was higher than 0.99.

In order to determine the detection and quantification limits (DL and QL), a standard solution of TTX and its analogues was prepared at a concentration near the expected detection limit and analyzed seven times. DL and QL were then calculated as 3.3 and 10 times the standard deviation of the replicate measurements of each analyte, respectively, divided by the slope of the respective calibration curve. The DLs of TTX, 11-deoxyTTX, 4,9-anhydroΤΤΧ, 4-epiΤΤΧ, 11-norTTX-6-ol, 5,11/6,11-dideoxyTTX and 5,6,11-trideoxyTTX were 0.5, 0.2, 3.5, 0.5, 0.5, 0.1 and 0.5 pg μL^−1^, respectively (translated to method detection limits of approximately 0.007, 0.003, 0.048, 0.007, 0.007, 0.001 and 0.007 μg g^−1^, respectively), while the respective QLs ranged from 0.3 to 10.5 pg μL^−1^.

To evaluate the efficiency of the extraction procedure, one tissue sample of *L. sceleratus* was subjected to three consecutive extraction cycles, and the extracts were separately analyzed. These results showed that the percentage of TTX in the first extraction was 93.5%, while only 6.5% of TTX was extracted in the next two cycles. Identical results were obtained for TTX-analogues.

With regard to hormones quantitation, calibration curves were prepared using a series of eight standard solutions in the range of concentrations expected for each target analyte in the real samples. Linear standard curves were obtained for testosterone and androstenedione over the concentration range of 0.003–5 pg µL^−1^ and 0.02–5 pg µL^−1^, respectively, while power–fit curves were generated for cortisol, β-estradiol and GnRH over the concentration ranges of 0.1–198 pg µL^−1^, 0.18–5 pg µL^−1^ and 2.6–166 pg µL^−1^, respectively. The regression coefficient R^2^ of the calibration curves was higher than 0.98 for all hormones investigated, with the exception of β-estradiol (*R*^2^ = 0.93). The detection and quantitation limits were calculated as described above for TTX. The DLs for GnRH, cortisol, β-estradiol, androstenedione and testosterone were 2.60, 0.05, 0.08, 0.01 and 0.002 pg µL^−1^, respectively (translated to method detection limits of approximately 0.02, 0.001, 0.001, 0.0001 and 0.00002 μg g^−1^, respectively), while the respective QLs were 7.8, 0.15, 0.24, 0.03 and 0.006 pg µL^−1^. The analytical recoveries of hormones were assessed by comparing the responses of samples that were spiked with the target analytes before and after extraction, and these were found to range from 84% to 98%.

Identification of TTX, TTX-analogues and hormones was accomplished by comparing the MS/MS spectral data (i.e., at least one quantifier and one qualifier MS/MS transition were acquired for each analyte), the response ratio of quantifier to qualifier MS/MS transitions (i.e., within ±20%) and the retention time (i.e., within ±0.1 min) of the analytes (as obtained from the analysis of standard solutions) with those of the LC-MS/MS chromatographic peaks. The measurements made in this study were not subjected to recovery correction. In addition, blank subtraction was not necessary as the target analytes were not detectable in blank samples.

### 3.4. Statistical Analysis

Prior to statistical analysis, non-detectable gonadal concentrations of some TTX-analogues and hormones were replaced by half of the detection limit (i.e., 1.75 pg µL^−1^ for 4,9-anhydroTTX, 1.30 pg µL^−1^ for GnRH, 0.04 pg µL^−1^ for β-estradiol and 0.005 pg µL^−1^ for androstenedione). The Pearson’s correlation matrix was calculated to identify any statistically significant linear relationships between tetrodotoxin, hormones and fish morphological characteristics. For the dichotomous categorical variable of sex (male = 0, female = 1), the point-biserial correlation coefficient was applied, which is mathematically equivalent to Pearson’s correlation. Subsequently, forward stepwise linear regression was carried out in order to identify the best models that can predict TTX concentration in *L. sceleratus* gonads using an optimally selected subset of variables. With this approach, the different variables are sequentially entered into the model, as far as the defined criterion of entry is satisfied (i.e., *p* < 0.05 for the partial F-test of each added variable). During the analysis, one gonad sample was identified as a multivariate outlier (i.e., Mahalanobis distance = 22.6 and Cook’s distance = 0.29, with both being three times higher than the respective average distance values) and it was excluded from the dataset. Moreover, since fish weight demonstrated a strong correlation with fish length (i.e., Pearson’s *r* = 0.95), the latter variable was excluded to avoid multicollinearity problems. All statistical analyses were performed using IBM SPSS Statistics for Windows (v22.0).

## 4. Conclusions

In the present study, different tissues of *L. sceleratus* specimens from southeastern Aegean Sea were analyzed and it was confirmed that liver and gonads are the most toxic parts of this species, with the concentration of tetrodotoxin and its analogues being almost an order of magnitude higher compared to skin and muscle tissues. In addition, TTX was more abundant in the ovaries than in testis of the specimens under investigation. This may be attributed to the sampling period, which corresponded to the spawning season for this species, as females have been reported to be highly toxic during that time of the year. Moreover, 78% of the muscle samples analyzed were regarded as toxic, according to the Japanese toxicity threshold (2.2 μg TTX eq g^−1^), thus verifying that *L. sceleratus* is highly poisonous and should not be consumed under any circumstances. The hormonal profile of *L. sceleratus* was investigated for the first time, and four steroid hormones, namely cortisol, testosterone, androstenedione and β-estradiol, as well as the gonadotropin-releasing hormone (GnRH), were detected in the gonads. Androstenedione was more abundant in female specimens, while GnRH was found at higher concentrations in males. Both of them are involved in the reproductive and territorial behavior of fish. A possible relationship between pufferfish biotoxins and the recorded hormones was examined, with β-estradiol being the only one that was positively correlated with TTX and its analogues. However, a stepwise regression analysis of the results demonstrated that gonadal TTX levels of *L. sceleratus* are better predicted using sex and weight as independent variables, rather than β-estradiol and weight. This implies that the toxin is more likely controlled by the general sex-related characteristics of the pufferfish rather than by specific hormone-controlled processes. Nevertheless, whatever variables were included in our models, these could explain less than two-thirds of the variation in TTX concentration. Further research is needed to unveil other hidden factors influencing the toxicity of this noxious pufferfish.

## Figures and Tables

**Figure 1 marinedrugs-21-00520-f001:**
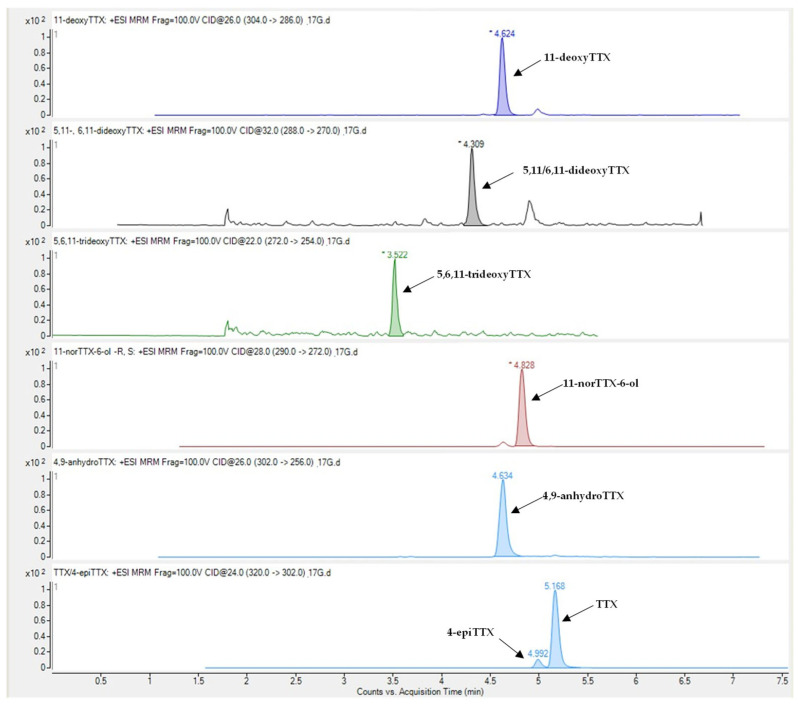
Representative LC-MS/MS chromatogram of TTX and its analogues in *Lagocephalus sceleratus* samples.

**Figure 2 marinedrugs-21-00520-f002:**
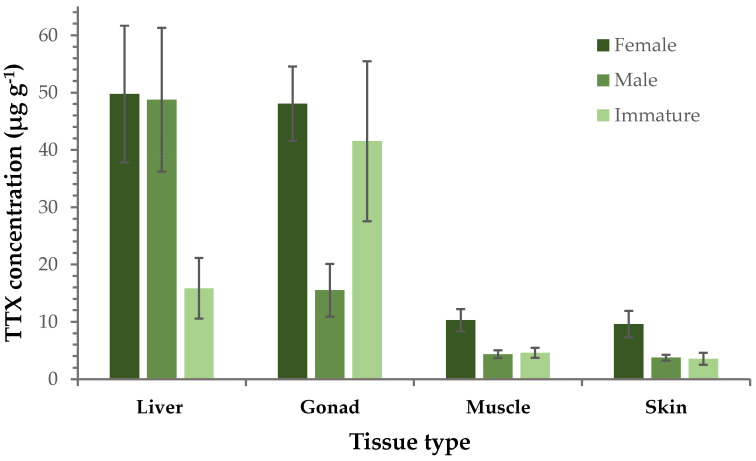
Average TTX concentration (μg g^−1^ ± SE) in each tissue type separately (liver, gonad, muscle, skin) from female, male and immature *L. sceleratus* specimens.

**Figure 3 marinedrugs-21-00520-f003:**
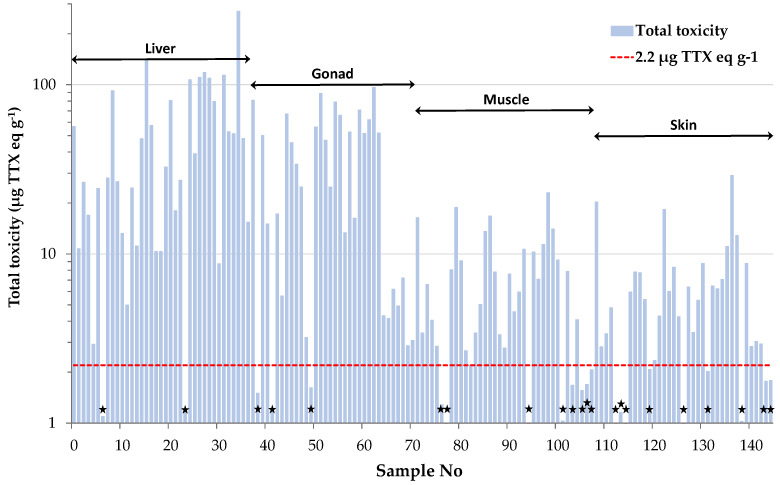
Toxicity assessment of *L. sceleratus* samples based on the Japanese safety consumption limit of 2.2 μg TTX eq g^−1^ (red dashed line). Non-toxic samples are marked with black asterisks.

**Figure 4 marinedrugs-21-00520-f004:**
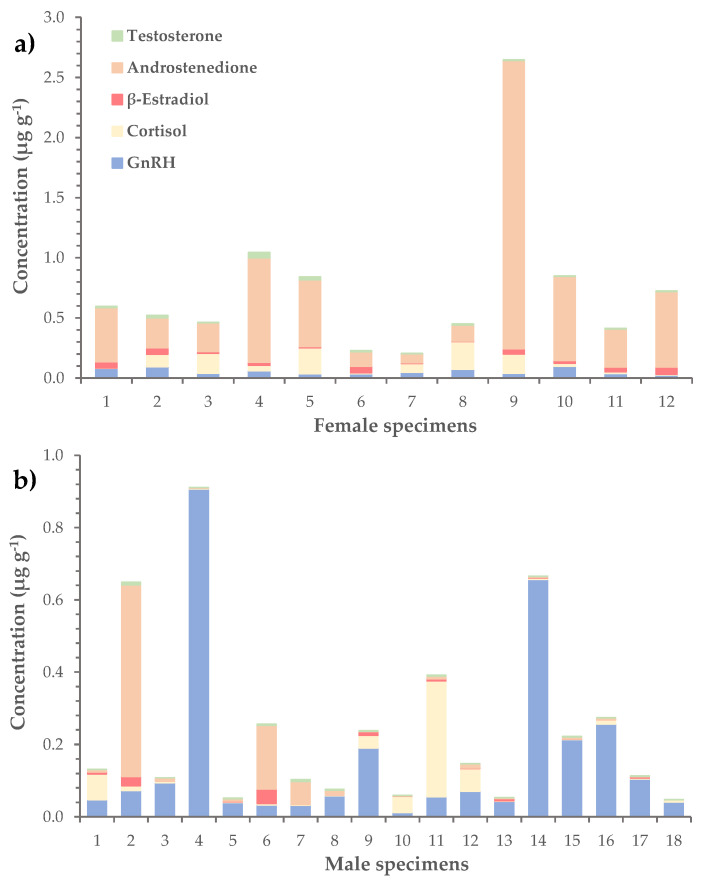
Distribution of GnRH and steroid hormones (i.e., cortisol, β-estradiol, androstenedione and testosterone) in the gonads of (**a**) female and (**b**) male *L. sceleratus* specimens.

**Table 1 marinedrugs-21-00520-t001:** Average concentration (in μg g^−1^; ± standard error), range of values (in μg g^−1^) and detection rate (%) of TTX and its analogues in different tissue types of 37 *L. sceleratus* specimens. Values below detection limit are reported as “n.d.”.

	Liver		Gonad		Muscle		Skin		Detection Rate
Analyte	Average	Range	Average	Range	Average	Range	Average	Range	
TTX	43.7 ± 7.7	1.1–239.8	30.1 ± 4.5	1.2–85.2	6.3 ± 0.9	0.2–21.7	5.6 ± 0.9	0.3–27.5	100%
4-epiTTX	5.4 ± 1.6	0.08–57.8	2.4 ± 0.5	0.02–11.1	0.3 ± 0.05	0.01–1.3	0.3 ± 0.07	0.04–2.4	100%
11-norTTX-6-ol	31.8 ± 5.6	0.07–115.6	18.2 ± 3.6	0.14–69.6	2.3 ± 0.4	0.02–9.9	2.3 ± 0.4	0.1–8.3	100%
4,9-anhydroTTX	3.7 ± 0.7	n.d.–24.1	2.1 ± 0.4	n.d.–9.1	0.2 ± 0.03	n.d.–0.8	0.2 ± 0.04	n.d.–1.2	93%
11-deoxyTTX	4.5 ± 0.8	0.01–20.1	1.6 ± 0.3	0.1–4.6	0.3 ± 0.04	0.01–1.0	0.3 ± 0.04	0.02–1.1	100%
5,11/6,11-dideoxyTTX	0.4 ± 0.1	0.002–2.1	0.2 ± 0.04	0.002–0.9	0.02 ± 0.004	n.d.–0.1	0.04 ± 0.01	0.01–0.2	99%

**Table 2 marinedrugs-21-00520-t002:** Total average toxicity (μg TTX eq g^−1^ ± SE) calculated for the different tissues of *L. sceleratus* using the relative potencies of TTX and TTX-analogues.

Tissue Type	Total Toxicity
Liver	51.03 ± 8.88
Gonad	33.99 ± 5.16
Muscle	6.81 ± 0.92
Skin	6.14 ± 0.97

**Table 3 marinedrugs-21-00520-t003:** Average concentration (in μg g^−1^; ± SE), range of values (in μg g^−1^) and detection rate (%) of the hormones found in *L. sceleratus* gonads. Values below detection limit are reported as “n.d.”.

Hormone	Average	Range	Detection Rate
GnRH	0.11 ± 0.03	n.d.–0.91	97%
Cortisol	0.06 ± 0.01	0.001–0.32	100%
β-Estradiol	0.020 ± 0.004	n.d.–0.06	76%
Androstenedione	0.23 ± 0.08	n.d.–2.40	97%
Testosterone	0.010 ± 0.002	0.0003–0.05	100%

**Table 4 marinedrugs-21-00520-t004:** Average concentration (μg g^−1^ ± SE) of the hormones detected in female, male and immature *L. sceleratus* specimens.

Hormone	Female	Male	Immature
GnRH	0.05 ± 0.01	0.2 ± 0.1	0.05 ± 0.01
Cortisol	0.10 ± 0.02	0.03 ± 0.02	0.06 ± 0.03
β-Estradiol	0.03 ± 0.01	0.01 ± 0.002	0.03 ± 0.02
Androstenedione	0.6 ± 0.2	0.05 ± 0.03	0.06 ± 0.04
Testosterone	0.010 ± 0.004	0.002 ± 0.0005	0.002 ± 0.0005

**Table 5 marinedrugs-21-00520-t005:** Pearson’s correlation coefficients (r) between hormones, morphological characteristics, TTX and TTX-analogues detected in *L. sceleratus* gonads.

	Weight	Length	Sex	TTX	4-epi TTX	11-nor TTX-6-ol	4,9-anhydro TTX	11-deoxy TTX	5,11/6,11- dideoxyTTX
Weight	−	0.949 **	−0.232	−0.476	−0.347	−0.365	−0.372	−0.468	−0.419
Length	0.949 **	−	−0.246	−0.388	−0.253	−0.266	−0.276	−0.374	−0.348
Sex	−0.232	−0.246	−	0.624 **	0.658 **	0.570 **	0.688 **	0.631 **	0.769 **
GnRH	0.397 *	0.374 *	−0.282	−0.321	−0.266	−0.289	−0.287	−0.313	−0.299
Cortisol	−0.178	−0.112	0.331	0.187	0.118	0.062	0.092	0.056	0.012
β−Estradiol	−0.325	−0.359	0.659 **	0.556 **	0.520 **	0.511 **	0.579 **	0.603 **	0.728 **
Androstenedione	−0.207	−0.265	0.539 **	0.132	0.11	0.163	0.145	0.193	0.212
Testosterone	−0.007	0.026	0.563 **	0.173	0.083	0.051	0.139	0.182	0.281

* *p* < 0.05, ** *p* < 0.01

**Table 6 marinedrugs-21-00520-t006:** Summary statistics and regression coefficients of the two models derived from forward stepwise multiple regression for the prediction of TTX concentration in *L. sceleratus* gonads. In the first model, all variables (i.e., pufferfish morphological characteristics and gonadal hormones levels) were assessed as potential predictors, whereas the sex variable was forced outside the second model.

Model No	R	R^2^	Standard Error of Estimate	*F*-Stat	*p*-Value	Variable	B ^a^	β ^b^	*t*-Test	*p*-Value	Part-R ^c^
1	0.787	0.619	16.8	21.8	*p* < 0.001	Constant	42.9	-	4.8	<0.001	-
						Sex	31.1	0.584	4.7	<0.001	0.572
						Weight	−0.013	−0.425	−3.4	0.002	−0.416
2	0.680	0.463	20.0	16.9	*p* < 0.001	Constant	38.5	-	4.1	<0.001	-
						β-Estradiol	600.9	0.487	3.5	0.002	0.464
						Weight	−0.010	−0.349	−2.5	0.019	−0.332

^a^ Unstandardized regression coefficient; ^b^ Standardized regression coefficient; ^c^ Semipartial correlation coefficient.

**Table 7 marinedrugs-21-00520-t007:** Number of collected *L. sceleratus* specimens from Rhodes Island, Greece categorized by sex (female, male, immature) and accompanied by measurements of total fish length (cm) and weight (g). The average values of the latter parameters are shown in parentheses.

Number of Specimens	Sex	Length	Weight
12	Female	42–65 (53 ± 8)	1030–3500 (1766 ± 761)
19	Male	49–72 (56 ± 7)	990–4300 (2138 ± 942)
6	Immature	23–45 (37 ± 8)	140–895 (568 ± 294)

## Data Availability

The data presented in this study are available on request from the corresponding author.

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
