# Peer review of "Assessing the Toxicity of Lagocephalus sceleratus Pufferfish from the Southeastern Aegean Sea and the Relationship of Tetrodotoxin with Gonadal Hormones"

_marinedrugs, 2023, doi:10.3390/md21100520_

Round 1
Reviewer 1 Report
The manuscript presents the toxicity of Lagocephalus sceleratus pufferfish from Southeastern Aegean Sea and the relationship of tetrodotoxin with gonadal hormones. The authors further present the identification of other sex-related characteristics are more important than specific hormone-regulated processes. Overall, the data is presented and discussed well. However, in part, some information is missing (e.g., method detection limits, and critical assay quality control measures).
In their revision, authors should consider the specific comments and errors specified below.
Detailed Comments:
Line 26 (Abstract)
There are two “from”, delete one.
Line 34 (Keywords)
The author should remove keywords that have appeared in the title, and also remove Eastern Mediterranean.
Line 42 (Introduction)
“It still poses a threat to local biodiversity, fisheries, and citizens.” Please add data and references for this sentence.
Line 48 “strong bite” of what? This sentence is incomplete
Lines 83-86 (Introduction)
Please add references.
Lines 367-391 (Materials and Methods) Section 3.1 TTX analysis and Section 3.2 Statistical analysis
Please add detection limits and quality control measures.
Table 7 (Materials and Methods: TTX analysis)
Add a column to the table for average weight and deviation. There is a significant difference in weight between male and female individuals of pufferfish, which may also affect the toxin content and distribution.
Line 373 (Materials and Methods: TTX analysis)
What gender are the three samples without gonads? Are these “immature”?A detailed explanation is required, otherwise, it affects the results of gender effects on toxin distribution.
Lines 383-384 (Materials and Methods: TTX analysis)
Please provide data support for a single extraction recovery of 93%. This can be added as an appendix.
Section 2.1. Concentration levels of TTX and related analogs
Please supplement the qualitative basis for homologs.
Lines 149-155 (Results and Discussion)
The total concentration of TTX is a conversion equivalent? Is it consistent with Figure 3? Authors need to double-check the calculations throughout the text, and if they are not uniform, they need to provide details in the methods section.
Lines 175-177 (Results and Discussion)
Please add references.
The information in Figure 2 shows that immature gonad has a high TTX concentration. Please add discussion.
Lines 220-224 (Results and Discussion)
The authors need to specify the sampling season of the samples.
Lines 213-238 (Results and Discussion)
The authors are summarizing the findings of others and not discussing them in the context of the content and objectives of the present study, so the authors need to delete and re-edit this section.
Line 246 (Results and Discussion)
“Quite recently” but the reference date is 2017
Line 251 (Results and Discussion)
“EFSA-limit” is misleading to readers and the authors need to consider changing it to the recommended limitation.
In Figure 3, the authors add the red 2.2ug marking line while comparing it to EFSA in the text. The intention expressed by the authors is not very clear.
The conclusion section is too long and provides a lot of unnecessary information. Need to rewrite.
The English language is fine.
Author Response
A detailed response is attached

Reviewer 2 Report
This is a well-designed study with interesting findings especially as regards the interrelation of TTXs contents with certain hormones in the L. sceleratus pufferfish species. There are, however, certain points needing some improvement in order to correct some points and make the manuscript more reader friendly.
General remarks:
- The quality parameters for both LC-MS/MS analyses (TTXs and hormones) should be reported, i.e. LOD, LOQ, linearity and recovery. LOD and LOQ are absolutely necessary to comment on the “non-detected” status of some of the samples.
- Throughout the manuscript, tables and figures: it would be preferable to report standard deviations (SD) instead of the standard error of the mean (SE) - it is a more representative measure. SE tends to mask the real variation because it is divided by number of samples (n).
Specific remarks:
Abstract:
- Page 1, lines 19-20: “… of L. sceleratus specimens from Rhodes island…”: would be good to mention how many samples and the time-span of sampling.
1. Introduction
- Page 1, lines 37-38: “the 13 pufferfish species recorded so far in the Mediterranean Sea [1]”: According to Ulman et al. 2021, it is eleven species that are established in the Mediterranean – maybe the authors should recheck this statement or use a different reference?
- Page 1, lines 40-43: “The widespread distribution…and citizens”: This is not only a local problem for Crete, it affects most of the Mediterranean countries, at least the eastern ones. The magnitude of the problem should be highlighted, treating it as a local issue probably diminishes its significance.
- Page 2, lines 83-86: A reference is needed to support this statement.
2. Results and Discussion
- Page 3, lines 103-104 “trideoxyTTX was detectable only in liver and gonad samples”: Please provide information on the method LOD and LOQ to facilitate readers’ comprehension. To what levels does “detectable” correspond?
- Page 4, line 141: “e.g. sex ratio, maturation stage, sampling area and season”: Additionally, sampling frequency, size of individuals, as well as number of samples could play an important role.
- Page 5, lines 171-172, Figure 2: The figure caption is misplaced – please move to the correct position.
- Page 7, line 244: Regulation (EC) No 854/2004 is no longer in force. The relevant text in question (dealing with the restrictions on trading pufferfish in the EU) is now included in the Commission Implementing Regulation (EU) 2019/627 (available at https://eur-lex.europa.eu/legal-content/EN/TXT/?qid=1676133711932&uri=CELEX%3A32019R0627). Please revise accordingly.
- Page 7, lines 255-256, Figure 3: It would be more comprehensive if the scaling was changed to have a better view under 100 - maybe split the y axis and display at a better zoom the lower values e.g. 0-100?.
- Page 7, line 267 and page 8, line 273 in Table 3: “was found below detection limit” and “Values below detection limit”: the LOD needs to be reported to understand these points.
3. Materials and Methods
- Page 11, lines 380-390: Please indicate the TTX standard used (manufacturer, purity or concentration etc).
- Page 12, lines 406-409: Please indicate the origin of the labelled hormones (manufacturer, purity or concentration etc).
4. Conclusions
- Page 13, lines 445-446: Once more the approach is too localized. It should be indicated that the L. sceleratus problem (and actually the Lessepsian pufferfish migrants one) is much more widespread to the Mediterranean, including the southern Aegean sea.
No significant issues detected - minor "polishing" required.
Author Response
A detailed response is attached

Reviewer 3 Report
I think that this is interesting paper reporting the results of investigation of the relationship between TTX and its analogs levels in organs of pufferfish and gonadal hormones. The data also seems solid.
I only found a small mistake to be revised.
page 1 line 22, abstract
“and5,11-6,11-dideoxyTTX” should be “and 5,11 /6,11-dideoxyTTX”
Author Response
A detailed response is attached

Reviewer 4 Report
This manuscript explores an important issue to understand the pufferfish that have invaded the Mediterranean in recent years. Pufferfish is a major public health concern due to its tetrodotoxin content. TTX is responsible for numerous - sometimes fatal - cases of human poisoning. This paper therefore makes an important contribution to a better understanding of the TTX in Lagocephalus sceleratus in the Mediterranean Sea. The work is well written, the methodology and analysis are consistent with the literature, and the discussion and conclusion are in line with the results. However, there are some minor errors in the manuscript and it needs to be corrected and annotated to make it more reader-friendly and consistent.
line 68-70; I think the introduction should mention the value set by the EFSA CONTAM panel (44 µg TTX/kg).
line 154-155; The fish samples in the cited study (reference number, 28) were collected from the Northeastern Mediterranean, not the Aegean Sea.
line 202; In the section "2.2. TTX-based toxicity of L. sceleratus" it may be problematic to discuss only the Japanese threshold value. The value set by the EFSA CONTAM Panel should also be used in this discussion. Otherwise, we cannot say that the analyzed pufferfish is non-toxic. Similarly, Figure 3 needs to be revised.
line 368; "3.1. TTX analysis" I suggest that LC/MS/MS conditions, column used, LOD, LOQ, etc. should be included in the text to make the article more reader-friendly. Also, what is the TTX standard used in the analysis? Please provide this information.
Author Response
A detailed response is attached
